# *Phytophthora* Species and Their Associations with Chaparral and Oak Woodland Vegetation in Southern California

**DOI:** 10.3390/jof11010033

**Published:** 2025-01-04

**Authors:** Sebastian N. Fajardo, Tyler B. Bourret, Susan J. Frankel, David M. Rizzo

**Affiliations:** 1Department of Plant Pathology, University of California, Davis, CA 95616, USA; tyler.bourret@usda.gov (T.B.B.); dmrizzo@ucdavis.edu (D.M.R.); 2Mycology and Nematology Genetic Diversity and Biology Laboratory, United States Department of Agriculture, Agricultural Research Service, Beltsville, MD 20705, USA; 3United States Forest Service, Pacific Southwest Research Station, Albany, CA 94710, USA; susan.frankel@usda.gov

**Keywords:** Mediterranean vegetation, oomycete, plant pathogen

## Abstract

Evidence of unintended introductions of *Phytophthora* species into native habitats has become increasingly prevalent in California. If not managed adequately, *Phytophthora* species can become devastating agricultural and forest plant pathogens. Additionally, California’s natural areas, characterized by a Mediterranean climate and dominated by chaparral (evergreen, drought-tolerant shrubs) and oak woodlands, lack sufficient baseline knowledge on *Phytophthora* biology and ecology, hindering effective management efforts. From 2018 to 2021, soil samples were collected from Angeles National Forest lands (Los Angeles County) with the objective of better understanding the diversity and distribution of *Phytophthora* species in Southern California. Forty sites were surveyed, and soil samples were taken from plant rhizospheres, riverbeds, and off-road vehicle tracks in chaparral and oak woodland areas. From these surveys, fourteen species of *Phytophthora* were detected, including *P. cactorum* (subclade 1a), *P. multivora* (subclade 2c), *P*. sp. *cadmea* (subclade 7a), *P*. taxon ‘oakpath’ (subclade 8e, first reported in this study), and several clade-6 species, including *P. crassamura*. *Phytophthora* species detected in rhizosphere soil were found underneath both symptomatic and asymptomatic plants and were most frequently associated with *Salvia mellifera*, *Quercus agrifolia*, and *Salix* sp. *Phytophthora* species were present in both chaparral and oak woodland areas and primarily in riparian areas, including detections in off-road tracks, trails, and riverbeds. Although these Mediterranean ecosystems are among the driest and most fire-prone areas in the United States, they harbor a large diversity of *Phytophthora* species, indicating a potential risk for disease for native Californian vegetation.

## 1. Introduction

The oomycete genus *Phytophthora*, family Peronosporaceae, kingdom Straminipila [1], includes several species that are significant plant pathogens worldwide [2,3,4,5,6]. In forests and natural ecosystems, invasive *Phytophthora* species can destabilize ecosystems with detrimental effects on biodiversity and on ecosystem services [7,8,9,10,11]. Since its detection in the mid-1990s in Northern California [12], *P. ramorum*, the causal agent of sudden oak death, is now present in 16 counties in California and has expanded its distribution to over 318.5 km^2^ in Oregon [11,13,14]. Western Australia has been severely affected by the “biological bulldozer” *P. cinnamomi*, which is capable of infecting 40% of the 5710 species of native flora [15,16].

Since the 1960s, there has been an exponential rise in devastating tree and shrub diseases caused by invasive *Phytophthora* species in natural ecosystems in Australia, Europe, and North America [17,18]. This is likely due to a combination of factors, including increased ecosystem disturbance associated with population growth and expanded plant cultivation and trade [19,20]. Studies focused on dispersal risk have shown that the inadvertent outplanting of infested nursery stock is a primary route for the unintended introduction of *Phytophthora* species into natural areas [7,21,22], with many recorded cases in native ecosystems in California [23,24,25,26,27,28]. Eradication efforts, however, are generally unsuccessful despite requiring labor-intensive management and incurring high associated costs, as well as increased surveys and improved diagnostic techniques [27,29,30].

California is one of the five main Mediterranean climate regions in the world, together with the European and North African Mediterranean Basin, Southern and Southwestern Australia, Cape Province in South Africa, and central coastal Chile [31,32,33,34,35,36]. These areas are biodiversity hot spots of endemic flora and fauna, containing about one sixth of the vascular plant species in the world in just 2.2% of the world’s land area [9,37,38,39]. Mediterranean climates support vegetation that provides essential ecosystem services such as climate amelioration, flood and erosion control, nutrient cycling, carbon sequestration, and wildlife habitat [40]. Furthermore, these areas are culturally significant to local indigenous people [41]. In California, a major representative of Mediterranean vegetation is the chaparral biome. These plant communities account for 9% of the vegetation land-cover, with a distribution that extends from the lower elevations of the coastal ranges and western slopes of the Sierra Nevada to the Transverse and Peninsular ranges in the Southern part of the state [42]. Chaparral comprises drought-tolerant evergreen sclerophyllous shrubs, with *Adenostoma* spp. (chamise and red-shanks, Rosaceae), *Arctostaphylos* spp. (manzanita, Ericaceae), and *Ceanothus* spp. (Rhamnaceae) being common representatives [43,44]. These species are prone to frequent wildfires but have evolved disturbance-resilient traits, such as resprouting, germination from persistent seed banks, and the development of deep tap roots [45,46,47]. Frequent fires and other human disturbances can severely degrade chaparral and other Mediterranean landscapes, often hindering their ability to recover completely [31,32,48,49]. To address this challenge, restoration activities are employed across various landscapes in California. These efforts may involve herbicide application, weeding, the outplanting of nursery-reared plants, seeding, or other treatments aimed at restoring impacted areas [32].

Restoration projects have historically relied on native plant container nursey stock as a primary source for materials. Many nurseries in California specialize in providing stock for restoration (CALSCAPE, https://calscape.org/plant_nursery.php (accessed on 3 December 2024)). However, pathogen surveys and sampling in nurseries have demonstrated that *Phytophthora* species are prevalent in horticultural settings, especially when sanitation best management practices are not adequately implemented [50,51,52,53,54]. *Phytophthora cactorum*, *P. crassamura*, *P. rosacearum*, and *P. ×cambivora* are the most frequently detected *Phytophthora* species in Californian restoration areas, with *P. cactorum* being amongst the most common in horticultural and forest settings across the state [28]. The risk of pathogen introductions is increasing due to intensifying restoration efforts in response to more wide-spread land degradation due to increased fire frequency and other natural and/or anthropogenic disturbances, and plantings aimed at reducing climate change.

The Angeles National Forest (ANF), in Southern California, USA, utilizes thousands of nursery-grown container native plants for the restoration of areas disturbed by fires, invasive plants, recreation activities (e.g., off-road vehicle use), and other special uses [34]. Surveys including pathogen testing determined that plants being grown in Southern California nurseries and outplanted at ANF restoration sites were contaminated with *Phytophthora* species [26]. However, the risk those introductions posed to restoration plantings and adjacent natural vegetation remained unknown. The main objective of the current study was to obtain an understanding of the current distribution patterns and diversity of *Phytophthora* species within ANF lands. This will serve as a baseline for comparison and indicate whether these introductions are damaging to adjacent vegetation and if the introduced *Phytophthora* species survive on outplanted native plants. This knowledge will aid in the monitoring and prevention of future pathogen introductions, enabling a more targeted approach to the development of best management practices and conservation efforts [55]. These goals prompted a National Fish and Wildlife Foundation (NFWF)-funded collaborative project between The United States Forest Service (USFS), ANF, and the Pacific Southwest Research Station (PSW), along with the University of California, Davis, Plant Pathology. The main objective of this initiative was to determine the distribution and diversity of *Phytophthora* species located on previously burned ANF lands that were being considered for restoration. This study presents the main outcomes and the potential risks associated with the presence of these *Phytophthora* species in the arid lands of Southern California.

## 2. Materials and Methods

### 2.1. Study Area and Sampling

The ANF is a 283,280 ha. protected area spanning the San Gabriel and Sierra Pelona Mountains managed by the USFS in Los Angeles County, USA (34°32′ N, 118°30′ W). The primary vegetation types are chamise chaparral and coastal sage scrub, dominated by shrub species including *Adenostoma fasciculatum* (chamise), *Eriogonum fasciculatum* (California buckwheat), *Eriodictyon crassifolium* (thickleaf yerba santa), and *Salvia mellifera* (black sage), which are often surrounded by native and exotic grasses (Figure 1A–C). In addition, larger shrubs can be found in this vegetation type, such as *Heteromeles arbutifolia*, *Rhus integrifolia*, and *Prunus ilicifolia*. In areas adjacent to natural waterways or rivers, *Quercus agrifolia* and the endangered *Berberis nevinii* grow together with other riparian plants, for example *Populus fremontii, Baccharis salicifolia*, and other *Salix* species (Figure 1D).

The ANF is characterized by having a Mediterranean climate, with average temperatures ranging from 8–20 °C in winter to 15–33 °C during the dry summer months, with a yearly average precipitation of 566 mm, which varies by elevation. Elevations range from 365 m to 3067 m above sea level, with Mt. Baldy Notch being among the areas with the highest elevation, where the average yearly snowfall is 334 cm.

Three past wildfires, the Copper (2002), Ranch (2006) and Sayre (2008) fires, have affected over 16,187 ha of the ANF [56]. Land disturbances can be observed within the three fire areas. Despite barriers and signage aimed to prohibit off-road vehicle use, vehicles tracks were observed, passing through grasslands and oak woodlands with several crossings or following through stream beds. Fencing has been erected to reduce grazing; however, evidence of cattle intrusion was commonly observed. Construction debris and soil movement were also observed in areas where powerlines or oil pipelines had been extended or repaired.

### 2.2. Site Selection and Sampling

Three areas across the ANF were evaluated in this study (Figure 2): (1) the northwestern area (NWA), characterized by montane chaparral; (2) the northeastern area (NEA) where oak woodlands and chamise chaparral dominate; and (3) the southwestern area (SWA), predominantly montane chaparral. Twenty-eight sites were selected based on ANF botanists’ criteria used for suitability for potential restoration. Twelve additional sites were added to the surveyed area to broaden the scope of the study and better represent chaparral and oak woodlands. Sites were surveyed for *Phytophthora* species over five field visits: May 2018, December 2018, March 2019, March 2020, and May 2021. Sampling in the NWA was discontinued in the last three field surveys as requested by the funder due to administrative changes.

Two of the NEA sites had been outplanted with restoration nursery stock prior to 2018, and restoration activities had been initiated in 2019 in three sites. Additionally, one restoration site in both the SWA and the NWA areas was sampled. Digital elevation models and flowlines for perennial and ephemeral streams were obtained from the United States Geological Services (USGS) databases, fire perimeters were obtained from the Fire and Resource Assessment Program (FRAP, CAL FIRE), and road and trails were obtained from a USFS database.

At each site, 10 to 15 soil samples were taken either from plant (rhizosphere)-associated soil or bulk soil (from off-road vehicle trails, river basins, or bare-ground). Plant species were selected for sampling based on their relative dominance at the site. Plant individuals were sampled if they presented typical *Phytophthora*-associated symptoms (dieback, wilting, or chlorosis). Asymptomatic plants were also sampled to determine the possibility of the presence of *Phytophthora* species without associated symptoms. The sampling scheme was based on previous surveys conducted in other regions of the world which aimed to unveil *Phytophthora* diversity [57,58,59,60]. Each sample consisted of approximately 1000 mL of soil collected from 1–30 cm below the soil line [61,62]. Roots were included in the sample when present. For each sample, GPS location, vegetation class (chaparral or oak woodland), sample category (upland or stream bed), and plant health were recorded. Plant health was assessed for all sampled trees and shrubs by a general rating, ranging from 1 (healthy), to 2 (50% or less affected canopy), 3 (75% of canopy affected), and 4 (dead). Samples were maintained in sealed, 1-gallon plastic bags in coolers with ice packs for transport to the Rizzo laboratory, UC Davis, for processing. Across all field visits, a total of 576 soil samples were collected among the three areas.

### 2.3. Baiting, Isolation, Identification, and Isolate Storage

*Phytophthora* baiting was conducted according to Erwin and Ribeiro [63]. A green D’Anjou pear and a rhododendron leaf were placed in each soil sample bag. Double-distilled water (ddH2O) was added to the sample bags until the water level was about 3 cm above the soil surface, ensuring both baits were in contact with the water. Baited soils were maintained at 20–22 °C with indirect natural light for 6–7 days. Once lesions appeared, pears were removed from the baiting bag and isolations were made from the lesions. Pear lesions were individually excised and submerged in CMA-PARP (modified from Erwin and Ribeiro [63]: 15 g/L corn meal agar; 0.025 g/L pentachloronitrobenzene; 0.25 g/L ampicillin; 0.01 g/L rifampicin; 0.01 g/L pimaricin). For rhododendron baits, leaves were surface-sterilized for 1 min in 5% bleach and rinsed with ddH2O. Twelve to fifteen 0.5 cm diameter leaf discs were excised from leaves and plated on CMA-PARP. Isolation plates were incubated in the dark at 18 °C for two weeks and checked periodically for growth. *Phytophthora*-like growth from pear lesions and rhododendron leaf disks were subcultured onto CMA-PARP and maintained in the dark at 20 °C. Isolates were morphotyped based on cultural morphology. Indistinguishable isolates from the same sample were “dereplicated”, and a representative isolate was selected.

The pure cultures of each isolate were transferred to a sterile 1 mL pea broth solution and grown at 20 °C for at least 48 h for DNA extraction [64]. A tuft of growing mycelium from the pea broth vials was removed with a sterile needle and placed in 100 μL of PrepMan Ultra kit (Thermo Fisher Scientific, Waltham, MA, USA), and DNA was extracted according to the manufacturer’s instructions. The oomycete-specific primer pair FRiz + ITS4TT was used to amplify the internal transcribed spacer (ITS) locus of the isolates according to previously reported conditions [64]. PCR products were prepared for Sanger sequencing with Exo-SAP IT (Thermo Fisher, Waltham, MA, USA) according to the manufacturer’s instructions, and sequencing was performed with PCR primers by the UC Davis College of Biological Sciences, DNA Sequencing Facility. Contigs and finished sequences were formed from the sequencing runs, as previously described [64].

ITS sequences were compared to those available in the GenBank nucleotide collection using BLAST searches (https://blast.ncbi.nlm.nih.gov/Blast.cgi (last accessed on 24 October 2024)). Positive determinations were made based on 100% matches to sequences from strains of verifiable identity, and additional loci were consulted for the remaining isolates [65]. For the long-term storage of representative isolates, 4–5-day pure cultures in 1/3 strength clarified V8 agar were transferred to sterile vials containing 1 mL water in duplicate and stored at 14 °C and 20 °C.

### 2.4. Phylogenetic Inference

Additional sequences were obtained for selected isolates to clarify ambiguous phylogenetic relationships for which ITS was not sufficient for species determination. The molecular determination of clade-7 and clade-8 isolates was, as described by Bourret et al. [66], based on the phylogenetic analysis of two single-locus datasets that were composed of ITS and mt cox1 (Appendix A). For clade-7 isolates, two strains produced ITS sequences that suggested an affinity to subclade 7a and a close relationship to *P*. *abietivora* but could not be determined as a previously described species. The ITS and mt cox1 datasets were created featuring verified members of subclade 7a using *P. sojae* and *P. cinnamomi* as outgroups and representatives of subclades 7b and 7c, respectively. For clade 8, the ITS sequence of a single strain suggested a phylogenetic placement in subclade 8e, with proximity to *P. marrasii*. For this, the ITS and mt. cox1 datasets were put together with verified members of clade 8, with *P. ilii* from clade 11 and *P. stricta* from clade 17 as the outgroups.

Phylogenetic analyses were conducted as described in Bourret et al. [23]. Datasets were aligned using MAFT L-ins-l [67] within AliView v1.28 [68]. Maximum likelihood trees were inferred by using IQTREE2 v2.3.6 [69]. TreeGraph2 v2.15.0 [70] was used to apply Bayesian posterior probabilities to maximum likelihood trees, which were later annotated with Inkscape (www.inkscape.org, accessed on 2 December 2024).

## 3. Results

### 3.1. Phytophthora Species Incidence

*Phytophthora* species were detected at 22 of the 40 sampled sites, encompassing all three areas. Of the 576 samples, 73 (~12%) were positive for *Phytophthora* (Appendix A), 308 were positive for *Pythium* spp. sensu lato (including *Globisporangium* spp., *Phytopythium* spp. and *Pythium* spp. s.s.), and 195 were negative for oomycetes. Positive *Phytophthora* samples were collected from 475 m to 1011 m above sea level, and across all 5 survey dates with a 5.3% *Phytophthora* positivity rate out of 56 samples in May 2018, 6.2% of 161 samples in December 2018, 18% of 199 in March 2019, 16% of 53 in March 2020, and 14% of 107 during May 2021. The GenBank accessions numbers of the ITS sequences of representative isolates of all fourteen species of *Phytophthora* and additional *P. cactorum* ITS haplotypes are given in Table 1.

### 3.2. Phytophthora Community Composition

From the 73 *Phytophthora*-positive samples, 87 *Phytophthora* isolates were obtained (Figure 3). The most frequently detected *Phytophthora* species were in clade 6b, *P. gonapodyides* (23 isolates matching strain P-1905, Gen Bank accession n° MK908981.1), followed by *P. crassamura* (14 isolates, strain CPHST BL 151 Gen bank accession n° MG865482.1), *P. megasperma* (three isolates, strain WPC: P6957, GU258767.1), and *P. chlamydospora* (three isolates, strain Ex-type CPHST BL 156, MG865471.1) from clade 6c (Table 1). In lesser frequency, isolates presenting two distinct ITS haplotypes of *P. cactorum* (clade 1a) were detected. Two *P. cactorum* isolates (haplotype CAC1) matched CBS voucher isolate 108.09, Gen bank accession N°KJ128036.1. The other three *P. cactorum* isolates (haplotype CAC3) differed by 2 bp from CAC1, with two polymorphisms in positions 616 and 717. All six isolates of *P. multivora* (clade 2c) differed by 1 bp from strain ex-type CPHST BL 104. The clade with most representatives was clade 6, with *P. inundata* (seven isolates matching strain CPHST BL 130, MG865517.1) and *P. rosacearum* (three isolates, differing by two bp from CBS:124696, GU259045.1) from subclade-6a, and *P. lacustris* (six isolates, differed one bp form ex-type P245 AF266793.2), *P. lacustris* x *riparia* (one isolate, matching strain SM15APR_ARG, MG696509.1) and *P. riparia* (nine isolates, differing by 1bp from strain CPHST ex-type BL 111, MG865583.1) from subclade 6d.

Three *Phytophthora* species that did not match any previously described species were also detected. Four isolates were recovered of *P*. sp. ‘pocumtuck’ (matching strain Rizzo Lab N°: SM08FEB_FLZ1, MG696351.1) from subclade-6b [71]. Additionally, ITS sequences from two isolates indicated a previously undescribed species closely related to *P. europaea* of subclade 7a. Upon further analysis, these two isolates were found to be 100% homologous to two strains detected in previous Rizzo Lab surveys of restoration areas in Santa Clara, County, CA (Rizzo Lab N°: SCVWD235, MG707801.1 and SCVWD621, MG707855.1). The two strains in the current study, plus the two strains from Santa Clara County, were given the provisional name of *P*. sp. *cadmea.* ITS phylogenies indicate that these strains cluster within subclade 7a and cannot be distinguished from *P. abietivora* (Figure 4). However, the mt. cox1 tree shows a phylogenetic distinction between *P.* sp. *cadmea* and most strains of *P. abietivora,* including the ex-type, as well as other species of subclade 7a (Figure 5). Lastly, a single record was made of a phylogenetically clade 8 strain. Phylogenetic inference of both ITS (Figure 6) and mt. cox1 (Figure 7) loci placed this isolate, introduced as a new provisional species, namely, *P.* taxon ‘oakpath’, within subclade 8e, with a distinct position from the only other species in the subclade, *P. marrasii*.

Thirteen of the fourteen *Phytophthora* species were detected in NEA sites, but *P. multivora* was only detected in SWA. Alongside *P. multivora,* SWA sites also yielded *P. cactorum*, *P. crassamura*, and *P. gonapodyides*. Only two *Phytophthora*-positive samples were recorded for the NWA, with both samples being positive for *P. crassamura*.

Twenty-two plant species were sampled from fourteen plant families in oak woodland areas, primarily *B. salicifolia* (Asteraceae), *Q. agrifolia* (Fagaceae), and *Populus fremontii* (Salicaceae). Less intensive sampling occurred from *Diplacus auricantus* (Phrymaceae) and *Hazardia squarrosa* (Asteraceae). Out of the 144 samples taken from upland oak woodland areas, 13% were positive for *Phytophthora* species: *P. cactorum*, *P. crassamura*, *P. gonapodyides*, *P. lacustris*, *P. lacustris* × *riparia*, *P. megasperma*, *P. riparia*, *P*. sp. *cadmea*, and *P.* sp. ‘pocumtuck’. The most common *Phytophthora*-positive associated hosts were *Q. agrifolia*, from which *P. cactorum*, *P*. *crassamura*, *P. gonapodyides*, and *P. riparia* were found in the rhizosphere soil, as well as *P. lacustris*, *P. lacustris* × *riparia*, and *P*. sp. ‘pocumtuck’, which were detected underneath *Salix* sp. individuals. These subclade-6b *Phytophthora* species were also found to be present in the rhizosphere soil of grasslands and from a *B. salicifolia* with mild crown thinning.

### 3.3. Phytophthora Species from Areas with Off-Highway Vehicle Tracks

Across all three areas, off-highway vehicle (OHV) tracks were observed in twenty sites. OHV tracks varied in terms of the severity of disturbance, which depended primarily on the vehicles used to make the tracks and the amount of use (motorbikes, heavy machinery, or trucks). In total, sixty-three samples were taken from OHV tracks, fifty-four from within chaparral, and nine from oak woodlands. These tracks had an 11% positivity-rate with seven *Phytophthora*-positive samples. *P. crassamura* was the most frequently isolated, followed by *P. gonapodyides*, *P*. sp. ‘pocumtuck’, *P. riparia*, and the only detection of *P*. taxon ‘oakpath’ (Figure 8H).

### 3.4. Association Between Phytophthora Incidence and Disease Symptoms

In total, 450 soil samples were taken from underneath plants rated for overall health; from 48 plant species, 24 plant families, 149 samples were taken from underneath plants rated with a “1” (appeared healthy), 127 were rated with a “2” (50% or less affected canopy), 78 were rate with a “3” (75% of canopy affected), and 96 were rated with a “4” (dead).

Twenty-nine samples, from thirteen plant species, were positive for *Phytophthora* species and were associated with plants rated from 1 to 4. Plants rated with a “1-healthy” had higher *Phytophthora* species diversity when compared with symptomatic plants. *P. cactorum*, *P. crassamura*, *P. gonapodyides*, *P. lacustris* x *riparia*, *P. multivora*, *P. riparia*, *P*. sp. *cadmea*, and *P*. sp. ‘pocumtuck’ were all found under asymptomatic individuals from seven sites, mostly in oak woodland areas. Asymptomatic *Q. agrifolia* from two sites harbored *P. crassamura* and *P. cactorum*, while other clade-6 species were found to be associated with *Salix* sp. In the SWA, *P. multivora* was isolated from two healthy-appearing individuals of *Toxicodendron diversilobum*.

The most common symptoms observed across sampled plant species were crown thinning and branch dieback, with some individuals presenting yellowing and leaf tip necrosis. From rhizosphere soil samples under symptomatic plants (nine plant species), *P. cactorum*, *P. crassamura*, *P. gonapodyides*, *P. inundata*, *P. lacustris*, *P. megasperma*, *P. multivora*, and *P. riparia* were detected. In total, 78 dead plants were sampled, from which four individuals were positive for *Phytophthora* species, *P. crassamura*, *P. multivora*, and *P. gonapodyides*. Symptomatic *Phytophthora*-associated plants were observed at twelve sites, eight from oak woodland areas and four from chaparral areas. *Phytophthora cactorum* was isolated underneath a *H. squarrosa*, with several leaves presenting tip necrosis and *E. crassifolium* with severe yellowing, especially on its bottom branches (Figure 8A). Additionally, *P. cactorum* was isolated underneath a *Q. agrifolia* with clear symptoms of dieback of the upper crown. *P. crassamura* was detected underneath a *B. salicifolia* with sparse branches and thinning and from two dead *S. mellifera* plants. *Phytophthora gonapodyides* was the only *Phytophthora* species found in a restoration basin across the three areas. This basin had previously been planted with *Q. agrifolia*, which had a dead crown with brown leaves still attached (Figure 8B). Adjacent to this restoration basin, *P. gonapodyides* was again isolated from a cluster of thinning *A*. *fasciculatum* plants (Figure 8E). At other sites, *P. gonapodyides* was isolated from *B. salicifolia* and *Q. agrifolia* with mild crown dieback. Other clade-6 species were also found underneath plants with mild symptoms, with the following host associations: *P. lacustris*-*Salix*. sp., *P. inundata*-*B. salicifolia*, *P. megasperma*-*D. aurantiacus*, and *P. riparia*-*Q. agrifolia* (Table 1). Additionally, *P. multivora* was isolated underneath dead *S. mellifera* and *E. crassifolium*, and it was also detected underneath *S. mellifera* and *E. fasciculatum* with some branch dieback.

## 4. Discussion

This survey represents the first general *Phytophthora* survey in Southern California natural areas, and the most recent of the few studies of *Phytophthora* diversity in Mediterranean regions [57,72,73,74,75,76,77]. Across the forty sites in both chaparral and oak woodlands areas, which included riparian areas and streams, fourteen *Phytophthora* species were detected. Considering the relatively low sampling intensity and the arid nature of the region, *Phytophthora* diversity was higher than expected. The three most frequently detected species in this survey, *P. gonapodyides*, *P. crassamura*, and *P. riparia*, were found to be associated with streams in both chaparral and oak woodlands. These species, in addition to *P. cactorum* and *P. multivora*, are also commonly encountered in forest and restoration scenarios in California [23,28,78] and in natural areas in other US states [71,79,80,81,82,83,84,85]

A comparable *Phytophthora* survey of Mediterranean-type plant communities in the National Park of La Maddalena archipelago (Northeast Sardinia, Italy) recovered nine *Phytophthora* species [57]. Of the *Phytophthora* species associated with the Italian maquis vegetation, two were found in the current ANF survey: *P. gonapodyides*, found in Sardinia waterways; and *P. crassamura*, detected in the rhizosphere soil of *Juniperus phoenicea* (Cupressaceae). Although different flora were sampled in the two surveys, a comparison of *Phytophthora*-positive plant families can be made. In the Anacardiaceae, *T. diversilobum* in the ANF chaparral areas and *Pistacia lentiscus* (mastic) from the La Maddalena archipelago were *Phytophthora*-positive, and *Rhus integrifolia* on the ANF was negative. Only *P. multivora* was found underneath *T*. *diversilobum*, and *P. gonapodyides*, *P. ornamentata*, and *P*. *bilorbang* were all associated with *P. lentiscus* in Sardinia. In Scanu et al.’s [57] study, *Phytophthora* species were also isolated from symptomatic and declining *Asparagus albus* (Asparagaceae), *J. oxycedrus* (Cupressaceae), and *Rhamnus alaternus* (Rhamnaceae). Species in the same plant families were sampled on the ANF, but no *Phytophthora* species were detected. These ANF plant species included *Hesperoyucca whipplei* (Asparagaceae), *Calocedrus decurrens* (Cupressaceae), and *Ceanothus* sp. (Rhamnaceae). Although, in the ANF, these species were *Phytophthora*-negative, their phylogenetic relationship could indicate potential host affinity for other *Phytophthora* species.

Five distinct ITS haplotypes of *P. cactorum* are present across California [66]. Of these five, two were detected among the *P. cactorum* isolates of the ANF survey: CAC1 and CAC3. CAC1 was found in a single site in an oak woodland upland area in NEA, underneath a mildly symptomatic *Q. agrifolia* tree and *H. squarrosa* shrub. CAC3 was found in the NEA and SWA, underneath a yellowing *Er. crassifolium* and a thinning *Salix* sp. In California, CAC3 is the most commonly encountered ITS haplotype, being present in restoration outplantings in both Northern and Southern California, and it is frequently detected in native plant nurseries [66]. CAC1 has a more limited distribution, being only found in natural areas and streams of California. ANF CAC3 isolates were found to belong to a larger worldwide lineage called the “apple–oak lineage” present in Czechia, Germany, Japan, The Netherlands, New Zealand, Russia, the UK, and the USA [23,28]. This suggests that the apple–oak lineage is moving locally and worldwide through horticulture [66]. Other isolates with the CAC1 ITS haplotype did not fit into any worldwide lineages in their analysis, and more information is needed to better understand the phylogeography of *P. cactorum* in California and in a worldwide context. The widespread distribution of *P. cactorum* across the ANF and the prevalence of certain ITS haplotypes raise concerns about its adaptability and ease of establishment in diverse environments. This adaptability is evident in *P. cactorum*’s wide host range, having been detected from economically and environmentally significant plants in horticultural, agricultural, forest, and restoration contexts [28,66,86,87].

In the ANF, *P. multivora* was only found in the SWA, detected in two upland chaparral sites with detections from rhizosphere samples of a healthy *T. diversilobum* and *E. fasciculatum* and from a dead *Er. crassifolium* and *S. mellifera*. In California, *P. multivora* is considered to have a more clonal population, with a single ITS haplotype encountered in nurseries, restorations, and in natural areas [28,88]. *P. multivora* is distributed globally in natural areas and streams. In Australia, it is associated with numerous dead and dying native hosts [73,89,90], and in Europe, it is associated with the plant trade and outplantings [59]. For example, in a small Italian nature reserve, *P. multivora* was associated with vegetation types that include plant genera similar to those encountered in the ANF, *Salix* spp., *Quercus* spp., and *Populus* spp. [75].

Of the fourteen *Phytophthora* species detected, *P. crassamura* was the only one that was found in all three sampled areas, in both vegetation types, in both upland areas and in streams, and on OHV tracks. *Phytophthora crassamura* was also found beneath asymptomatic and symptomatic plants from five different plants species, including a dead *S. mellifera*. *Phytophthora crassamura*, which is among the most common *Phytophthora* species encountered in California, with strong associations with agriculture and restoration areas and to a lesser extent with nurseries [28]. Reviewing historical records, considering the recent description of *P*. *crassamura* from maquis vegetation in Italy and with the possibility of previous *P. megasperma* isolates having been misidentified [28,57], *P. crassamura* may have been in the US for at least 100 years [28]. Sims et al. [91] conducted a phenotypic and genotypic analysis, using cytochrome c oxidase subunit 1 (cox1) sequences, oogonia size, and mefenoxam resistance variation, and concluded that *P. crassamura* lineages found in Northern California restoration areas had originally come from nurseries and were being spread into wildlands. Follow-up analysis will be conducted to determine the lineage of the ANF *P. crassamura* isolates and potential source of introduction.

Eleven out of the fourteen species detected (79%) and 84% of the total *Phytophthora* isolates recovered from the ANF survey were in clade 6. The majority belonged to clade-6b and were more frequent in streams and oak woodlands. Except for *P. crassamura*, most clade-6 species were associated with healthy or asymptomatic plants. Only three plants from which *P*. *gonapodyides* was isolated showed symptoms of crown thinning. Only one was dead: a *Q. agrifolia* plant from a restoration plant basin. In California, clade-6 species are common in natural areas and streams, and many are potentially indigenous based on high levels of intraspecific diversity [23,28]. Clade-6 species also are found in streams and riparian areas in many areas of North America and Europe [75,81,92]. They are mostly associated with a saprophytic lifestyle [79,93]. However, when inoculated or present in disease conducive conditions, these species can initiate disease and damage trees [61,81,92,93,94,95,96,97]. Globally clade-6 species such as *P. gonapodyides* and *P. inundata* have been commonly detected in Mediterranean areas [57,98,99] and in other parts of California [24]. This predominant aquatic and saprophytic lifestyle in natural areas needs to be further investigated to understand the survival and potential threat these species pose in these dry and arid areas.

The subclade 7a *Phytophthora* sp. *cadmea* isolates yielded in Rizzo Lab surveys revealed that this species is associated with natural and restoration areas of both Northern and Southern California. The closely related subclade 7a *Phytophthora*, *P. abietivora*, is associated with Christmas tree nurseries in North America [28,100]. Both species’ association with nurseries and/or restoration outplantings suggests that human activities are, in part, responsible for their present distributions, which may be of interest to regulators both locally and internationally. Additionally, the lack of sequences identified as *P. abietivora* covering the 5′ [101] portion of the cox1 locus and the low phylogenetic resolution of ITS in subclade 7a both likely contributed to the large number of *P. abietivora* accessions deposited as *P. europaea*, potentially underestimating the actual distribution of these species in North America. Furthermore, the intraspecific cox1 diversity observed among North American isolates of both *P. abietivora* and *P*. sp. *cadmea* suggests that these two species are native to the continent.

The sole representative of clade 8 in this current survey, *P.* taxon ‘oakpath’, was isolated from a bulk composite sample collected from a trail in an oak forest near a chaparral dominated area. Phylogenetically, the resolution obtained from ITS and cox1 loci clustered this species within subclade 8e and separated this isolate from *P. marrasii*. Recently identified in 2022, *P. marrasii* was associated with the decline of artichoke plants in Italy [102]. This discovery, along with reports of *P. syringae* and *P. cryptogea* in other Mediterranean regions of the world [28,57,74,75,89], suggests that clade 8 may harbor more species associated with drier climates than previously detected, increasing the risk in natural and agricultural scenarios. As seen in refs. [103,104,105,106], new understandings of Phytophthora ecology and biology arise from analyzing newly described species, analyzing new host associations, and conducting phylogeographic analyses of populations within previously described clades.

This survey was performed in a variety of ecological niches with the goal of determining the distribution and ecology of *Phytophthora* species. Streams and oak woodland areas had the highest diversity of *Phytophthora* species, with clade-6 most prevalent. In drier areas, the upland chaparral, homothallic species *P. cactorum*, *P. multivora*, and *P. crassamura* were more common. These species can readily produce thick-walled oospores, allowing them to survive adverse environmental conditions. It is unclear why *P. multivora* had such a narrow distribution in the ANF compared to other homothallic species, being limited to a few sites and only in the SWA. This could be due to many factors. For example, waterways in SWA are fed by the Upper Los Angeles River watershed, while in the NEA, the Bouquet Canyon and Upper Santa Clara are the dominant watersheds, indicating potentially different introduction points and/or dispersal patterns through the various waterways. Further sampling is required to determine if this corresponds to a recent introduction into the area and to assess the real distribution of this species in the ANF and in other areas of Southern California.

The widespread distribution of many *Phytophthora* species in the ANF is of concern as the nature of their presence is still unknown. Periodic or seasonal *Phytophthora* sampling is recommended for monitoring potential undetected *Phytophthora* species introductions. Detection of *Phytophthora* species in OHV tracks is also of concern, indicating a potential route of dispersal through human-associated soil movement [15]. Similarly, the monitoring of waterways is also recommended to assess a baseline of *Phytophthora* diversity and because waterways are potential vectors for new introductions. Despite the absence of positive detections in established restoration areas sampled in the current ANF study, caution and monitoring should be prioritized by nursery and restoration managers to prevent introductions as *Phytophthora* species were found to be common in California native plant nurseries [25].

## 5. Conclusions

In recent decades, interest in *Phytophthora* sampling has increased in areas historically considered unfavorable for the pathogen’s dispersal and pathogenicity, including Mediterranean plant communities [57,74,76,104,107]. These plant communities have unique ecological characteristics and support both a large diversity of flora and fauna and large human populations, making them more vulnerable to ever-increasing disturbances [31,32,108]. *Phytophthora* species in the ANF were found to be more widely distributed than anticipated, not only in dry and wet streams but also in seasonally dry uplands areas. Notably, homothallic species were more prevalent in upland habitats. This emphasizes the role these areas may have in the distribution of *Phytophthora* species and the importance of species-specific functional traits in influencing dispersal and establishment [109,110]. Soil compaction and poor drainage conditions created by OHV tracks can potentially create ideal soil conditions for proliferation and spread of *Phytophthora* species [29]. This highlights the importance of well drained soils on roads and planted areas to reduce the chances of facilitating *Phytophthora* conducive conditions. *Phytophthora* species were encountered underneath both symptomatic and asymptomatic plants, with limited detections in restoration sites, leaving the potential of overlooking *Phytophthora* infections in highly susceptible areas. Furthermore, the use of *Phytophthora* baiting methodologies carries the risk of false negatives, potentially overlooking species present in a certain sample [111,112]. To address this limitation, continued monitoring and repeated sampling are crucial, incorporating higher resolution detection techniques such as metabarcoding to complement traditional baiting methods [113]. This study narrowed the current knowledge gap of *Phytophthora* diversity in areas of Southern California, establishing new host associations, first detections, and the discovery of novel *Phytophthora* taxa. However, further research is needed to understand the threats these *Phytophthora* species pose to the local and native flora and to develop effective strategies for their prevention and management. These should include expanded surveys within the region and pathogenicity tests on common chaparral species.

## Figures and Tables

**Figure 1 jof-11-00033-f001:**
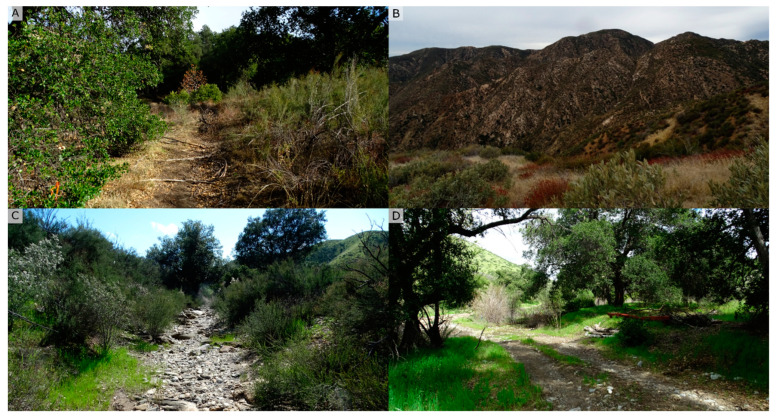
Examples of areas of the Angeles National Forest from which soil samples were taken to determine the presence of *Phytophthora* species. These include transition areas between chaparral and oak woodland areas (**A**), montane chaparral areas (**B**), rip riparian areas with dry and wet riverbeds (**C**), and oak woodlands (**D**). *Adenostoma* fasciculatum, *Eriodictyon crassifolium*, *Salvia mellifera*, and *Quercus agrifolia* were the most sampled native plant species in these areas.

**Figure 2 jof-11-00033-f002:**
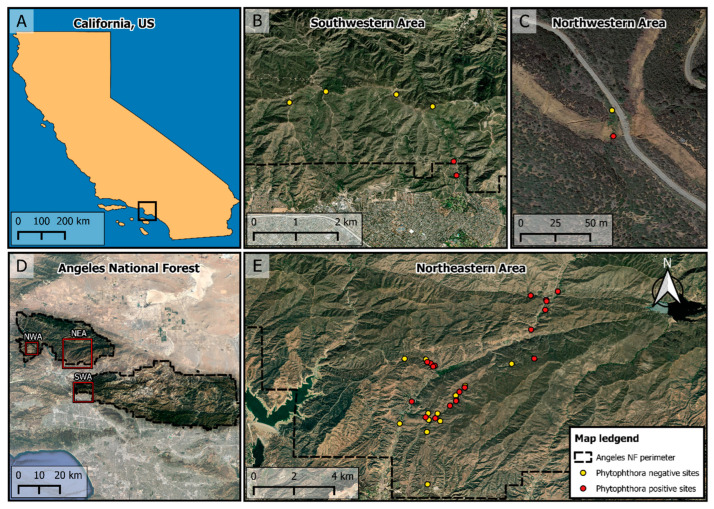
Location of the 40 sites sampled in the Angeles National Forest (ANF). Three main areas were sampled: the southwestern (SWA, **B**) area, with six sites; the northwestern (NWA, **C**) area, with two sites; and the northeastern (NEA, **E**) area, with 32 sites. Red dots indicate *Phytophthora*-positive sites, and yellow indicate *Phytophthora*-negative sites (**B**,**C**,**E**). Map (**A**) displays the location of the ANF in Southern California, with black box indicating Los Angeles County, and map (**D**) shows the general location of the sampling areas.

**Figure 3 jof-11-00033-f003:**
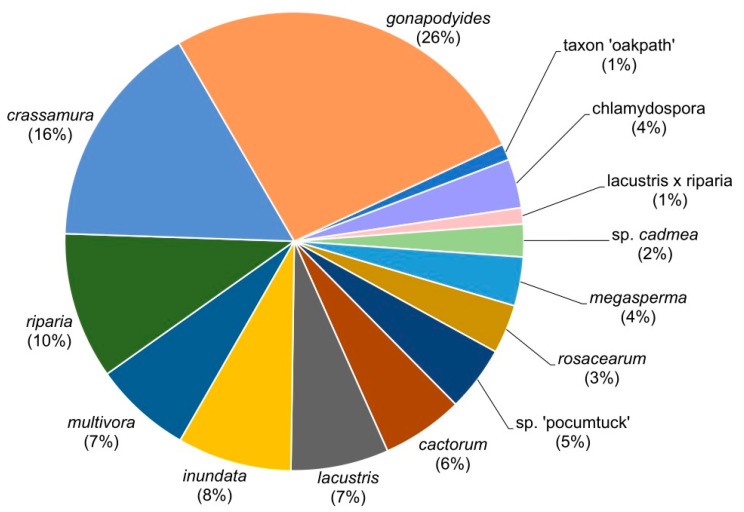
Diversity and frequency of *Phytophthora* taxa isolated from surveys of burnt areas of the Angeles National Forest from 2018 through 2021. Upland areas and streams beds were sampled in chaparral and oak woodlands, including restoration areas. Multiple isolates of a *Phytophthora* taxon from the same sample were considered as one record.

**Figure 4 jof-11-00033-f004:**
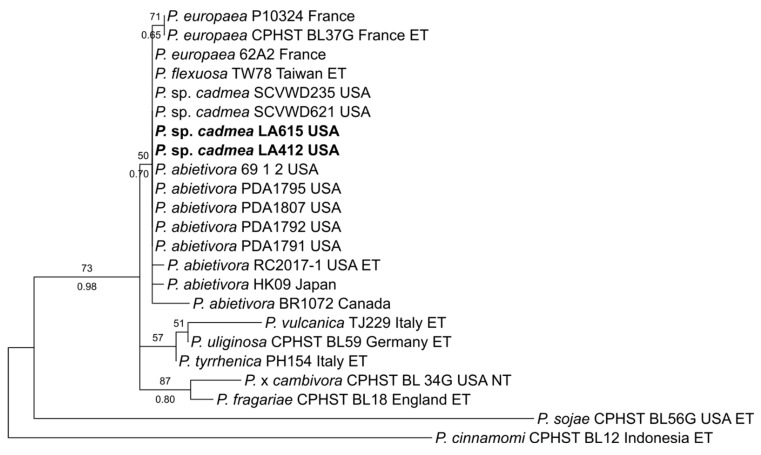
Maximum likelihood tree of *Phytophthora* subclade 7a. The tree is inferred with IQ-TREE 2 from single-locus ITS rDNA alignment. Support values above the branches are ultrafast bootstrap approximations ≥ 50, and those below are posterior probabilities ≥ 0.90, according to an analysis with MrBayes v2.3.7a. The isolates in bold are from this study.

**Figure 5 jof-11-00033-f005:**
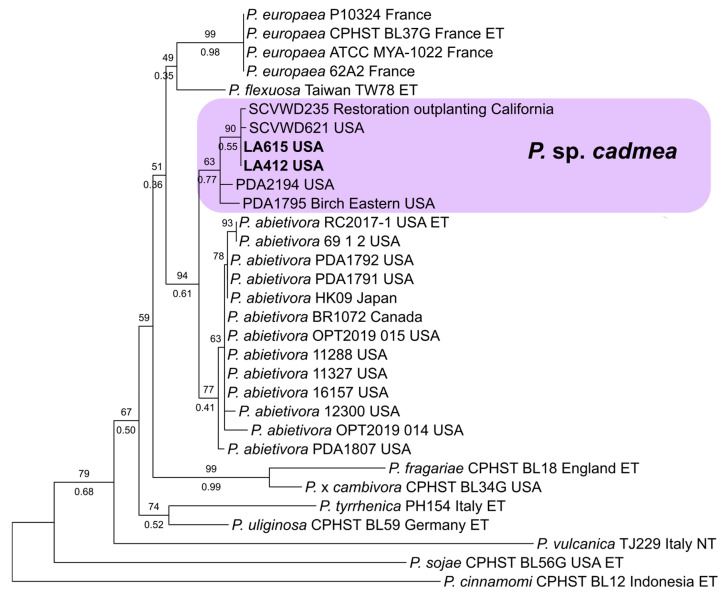
Maximum likelihood tree of *Phytophthora* subclade 7a. The tree is inferred with IQ-TREE 2 from a mitochondrial cox1 alignment. Support values above the branches are ultrafast bootstrap approximations ≥ 50, and those below are posterior probabilities ≥ 0.90, according to an analysis with MrBayes v2.3.7a. Isolates in bold are from this study. Isolates PDA2194 and PDA1795 were uploaded as *P. abietivora* and are included in the original description of the species.

**Figure 6 jof-11-00033-f006:**
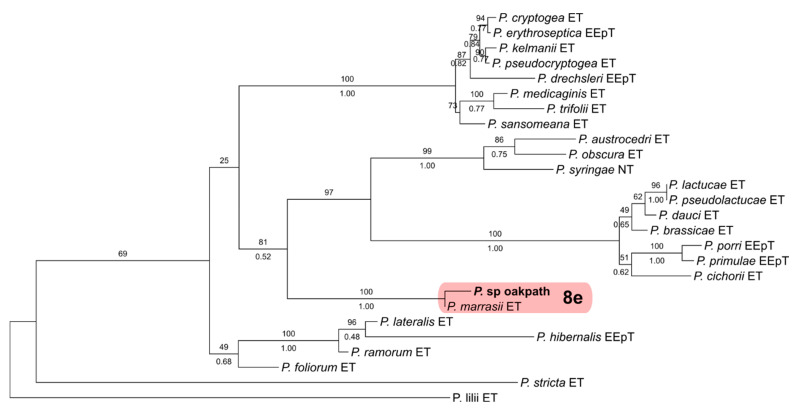
Maximum likelihood tree of *Phytophthora* clade 8. The tree is inferred with IQ-TREE 2 from single-locus ITS rDNA alignment. Support values above the branches are ultrafast bootstrap approximations ≥ 50, and those below are posterior probabilities ≥ 0.90, according to an analysis with MrBayes. The isolate in bold is from this study.

**Figure 7 jof-11-00033-f007:**
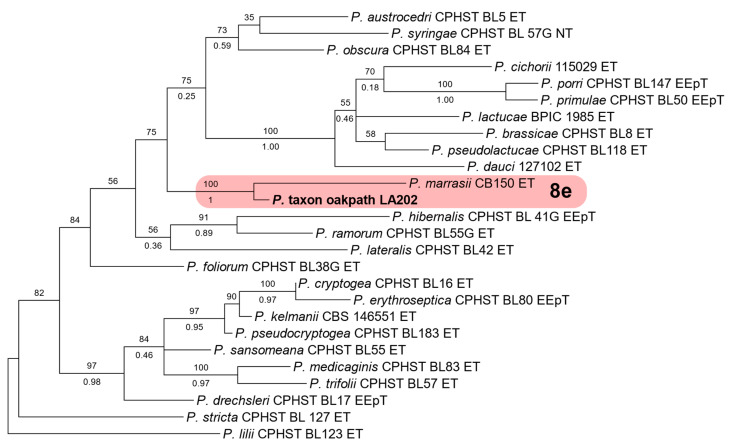
Maximum likelihood tree of *Phytophthora* clade 8. The tree is inferred with IQ-TREE 2 from a mitochondrial cox1 alignment. Support values above branches are ultrafast bootstrap approximations ≥ 50, and those below are posterior probabilities ≥ 0.90, according to an analysis with MrBayes v2.3.7a. The isolate in bold is from this study.

**Figure 8 jof-11-00033-f008:**
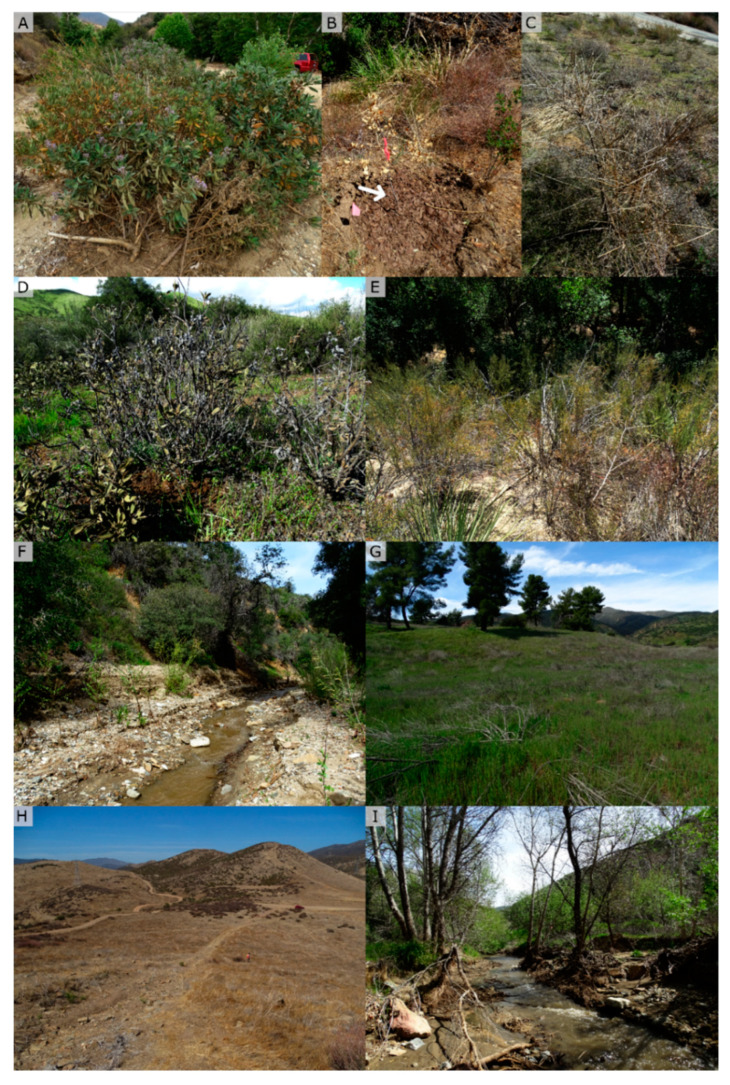
*Phytophthora*-positive plants and streams in fire areas of the Angeles National Forest: (**A**) *Eriodictyon crassifolium* positive for *P. cactorum*; (**B**) white arrow indicates dead *Quercus agrifolia* restoration plant positive for *P. gonapodyides*; (**C**) dead *Salvia mellifera* positive for *P. multivora*; (**D**) *E. crassifolium* with mild symptoms of crown thinning, close to a *P. crassamura*-positive stream; (**E**) thinning *Adenostoma fasciculatum* from which *P. gonapodyides* was isolated; (**F**–**I**) sites positive for *P. crassamura* and other clade-6 *Phytophthora* species.

**Table 1 jof-11-00033-t001:** *Phytophthora* species isolated from the Angeles National Forest, including sample source or host, sample type, and a representative isolate of each *Phytophthora* species.

Phytophthora Species	Sub Clade	No. of Isolates	Vegetation Type	Sample	Source/Host	ANF Area
*cactorum* (CAC3)	1a	3	Oak woodland	Rhizosphere	*Quercus agrifolia*, *Hazardia squarrosa*	NEA
*cactorum* (CAC1)	1a	2	Chaparral	Rhizosphere	*Salix* sp., *Eriodictyon crassifolium*	NEA, SWA
*chlamydospora*	6b	3	Chaparral	Bulk	Riverbed	NEA, SWA
*crassamura*	6b	14	Chaparral, Oak woodland	Rhizosphere, Bulk	*Artemisia californica*, *B. salicifolia*, Grass ^1^, OHV tracks ^2^, *Q. agrifolia*, *Salvia mellifera*, Riverbed ^3^,	NEA, SWA, NWA
*gonapodyides*	6b	23	Chaparral, Oak woodland	Rhizosphere, Bulk	*Adenostoma fasciculatum*, *B. salicifolia*, Grass, OHV tracks, *Populus fremontii*, *Q. agrifolia*, Riverbed	NEA, SWA
*inundata*	6a	7	Chaparral, Oak woodland	Rhizosphere, Bulk	*B. salicifolia*, Riverbed,	NEA
*lacustris*	6d	6	Chaparral, Oak woodland	Bulk	Riverbed, *Salix* sp.	NEA
*lacustris x riparia*	6d	1	Oak woodland	Rhizosphere, Bulk	*B. salicifolia*, Riverbed	NEA
*megasperma*	6b	3	Oak woodland	Rhizosphere, Bulk	*Diplacus auricantus*, Riverbed	NEA
*multivora*	2c	6	Chaparral	Rhizosphere, Bulk	*E. crassifolium*, *Eriogonum fasciculatum*, Riverbed, *S. mellifera*, *Toxicodendron diversilobum*	SWA
*riparia*	6d	9	Oak woodland	Rhizosphere, Bulk	Grass, OHV, *Q. agrifolia*, Riverbed, *Salix* sp.	NEA
*rosacearum*	6a	3	Chaparral	Rhizosphere, Bulk	Grass, Riverbed	NEA
sp. *cadmea*	7a	2	Chaparral, Oak woodland	Rhizosphere, Bulk	Grass, Riverbed	NEA
sp. ‘pocumtuck’	6b	4	Chaparral, Oak woodland	Rhizosphere, Bulk	*B. salicifolia*, OHV, Riverbed	NEA
taxon ‘oakpath’	8e	1	Oak woodland	Bulk	OHV	NEA

^1^ Native and exotic grasses (*Bromus* spp., *Avena* spp.). ^2^ Tracks left by off-highway vehicles (OHV). ^3^ Areas with evidence that water passes through, ephemerally or perennially.

## Data Availability

The original contributions presented in this study are included in the article/Appendix A. Further inquiries can be directed to the corresponding author.

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
