# Peer review of "Phytophthora* Species and Their Associations with Chaparral and Oak Woodland Vegetation in Southern California"

_jof, 2025, doi:10.3390/jof11010033_

Round 1
Reviewer 1 Report
The manuscript provides a comprehensive analysis of the Phytophthora species in Southern California, the tree and shrub diseases they cause, and their relationship with host plants. This study is significant, as it addresses the key link between understanding the increasing invasion of Aspergillus species and the interactions between soil and native plant communities. A relatively robust method has been provided through soil sampling analysis, lesion isolation, quantitative sequencing, and phylogenetic inference, but there are still areas that need to be modified. I only have few comments for the MS.
1.The objectives of the study are not explicitly stated. The abstract should clearly mention the specific goals, such as investigating the diversity of Phytophthora species, understanding the transmission pathways of Phytophthora species, or clarifying the mechanisms of infection.
2.The abstract lacks a clear introduction to determine the background and importance of studying Phytophthora species and their associations with chaparral vegetation in Southern California. It should briefly outline why this research is significant, such as the disease characteristics and potential risks of the Phytophthora species.
3.The manuscript indicates that 28 locations were evaluated in three regions of ANF. 7 to 10 soil samples were selected from each region, which is usually beneficial for statistical analysis. However, it is currently unclear whether these treatments were randomly assigned in these surveys or if there were systematic biases in how treatments were applied. Proper randomization and replication are crucial to ensure the reliability and generalizability of experimental results.
4.In Section 2.2, clarify whether the soil sampling depth of 1-30cm is based on previous research or specific to the root structure of sample categories.
5.I could not find any latest reference citations (from 2022,2023 2024) and cited literature in generally old. The authors must include latest articles related to the study in the discussion section and improve this section. Only one reference (83) is from 2023 that is on The Devastating Oomycete Phytopathogen.
6.Conclusion: this section is like summary not conclusions. authors should add clear 2-3 conclusions supported with results/research findings. Also mention the research limitations, not just the future research ideas.
7.In the iThenticate report provided, the number of repeated wording in this manuscript is too high. The author need make very systematic revisions after checking for duplicates.
Author Response
Comment 1: The objectives of the study are not explicitly stated. The abstract should clearly mention the specific goals, such as investigating the diversity of Phytophthora species, understanding the transmission pathways of Phytophthora species, or clarifying the mechanisms of infection.
Response 1: Thank you for your comment. To clarify, main objectives are now explicitly stated in the abstract and main body of the manuscript in lines 15-16 and 88-90, respectively.
Comment 2: The abstract lacks a clear introduction to determine the background and importance of studying Phytophthora species and their associations with chaparral vegetation in Southern California. It should briefly outline why this research is significant, such as the disease characteristics and potential risks of the Phytophthora species.
Response 2: We agree with your comment. Background and importance was incorporated in the abstract in lines 11-15.
Comments 3: The manuscript indicates that 28 locations were evaluated in three regions of ANF. 7 to 10 soil samples were selected from each region, which is usually beneficial for statistical analysis. However, it is currently unclear whether these treatments were randomly assigned in these surveys or if there were systematic biases in how treatments were applied. Proper randomization and replication are crucial to ensure the reliability and generalizability of experimental results.
Response 3:
In total 40 locations were sampled, taking between 10 to 15 samples at each site, thus a total of 576 soil samples. Since it corresponded to an exploratory survey, with the main objective of determining Phytophthora diversity, sampling was biased towards plants that presented typical Phytophthora-associated symptoms (dieback, wilting or chlorosis). Same approach can be seen in other Phytophthora surveys conducted in different regions of the world (Scanu et al. 2015, Jung et al. 2018, Jung et al. 2020, and Jung et al. 2017.
Comments 4: In Section 2.2, clarify whether the soil sampling depth of 1-30cm is based on previous research or specific to the root structure of sample categories.
Response 4: Thank you for your comment. This method is based on basic Phytophthora field sampling protocols, details can be found in Jung et al 2009 and Perez-Sierra et al. 2022, as referenced in the article in line 162.
Comments 5: I could not find any latest reference citations (from 2022,2023 2024) and cited literature in generally old. The authors must include latest articles related to the study in the discussion section and improve this section. Only one reference (83) is from 2023 that is on The Devastating Oomycete Phytopathogen.
Response 5: Thank you for pointing this out. We currently have in the manuscript 21 references that are after 2020: Perrine-walker, 2020; Ivanov et al. 2021; Cobb et al. 2020; LeBoldus et al. 2022; Molnar et. al 2020; Bourret et al. 2023; Swiecki et al. 2021; Bourret et al. 2023; Frankel et al. 2020; Meyer et al. 2021; Guarnaccia et al. 2021; Abad et al. 2023; Bourret et al. 2022; Minh et al. 2020; Riolo et al. 2020; Deidda et al. 2024; Chen et al. 2023; Tsykun et al. 2022; Jung et al. 2020; Bregant et al. 2021; Underwood et al. 2020.
After your comment we decided to add 10 additional references 2022-2024: Brasier et al. 2022; Horta-Jung et al. 2024; Jung et al 2024; Mullet et al. 2024; Chen et al. 2022, Mullet et al. 2023; Swiecki et al. 2024, Caballol et al. 2024, Sarker et al. 2023, Sarker et al. 2023b
Comments 6: Conclusion: this section is like summary not conclusions. authors should add clear 2-3 conclusions supported with results/research findings. Also mention the research limitations, not just the future research ideas.
Response 6: Thank you for highlighting this. For this we have added conclusions supported with the findings in the current study.
Comment 7:In the iThenticate report provided, the number of repeated wording in this manuscript is too high. The author need make very systematic revisions after checking for duplicates.
Response 7: Thank you for your feedback on the English language. This manuscript has been carefully reviewed by multiple native English-speaking co-authors, ensuring the grammar and overall English are of high quality. We believe the current English is appropriate for scientific publication. No other reviewer asked for a revision of the wording.
Reviewer 2 Report
Dear authors,
Congratulations for the work submitted and for the results obtained and so well discussed.
The work submitted is an original survey searching for Phytophthora species associated to Mediterranean forests in California. The research presented in the manuscript is linked with other works from the same research team, who are very experienced with the genus Phytophthora and the location of the surveys. As far as I know this is the first time that this work and the results are availale for publishing. Similar works have been made for other type of forests and regions, but the manuscript include original and interesting information for Mediterranean forest and nurseries managers.
The manuscript is well organised, the introduction is updated and is adequate, as well as the methodology, which is consistent with the discussion of the results.
From my point of view the manuscript is acceptable after minor changes, and I have some comments and questions to the authors:
1- The title could be modified in view of the origin of the samples as: “Phytophthora species and their associations with chaparral and oak woodland in Southern California”.
2- Line 29: Please, consider to change “among the most” by “include species”.
3- Line 39: Check ref 17 (is related to Vietnam (Asia)).
4- Line 49: I wonder if is necessary to cite 6 references (31-36) for the information included in the paragraph.
5- Line 69: Check if the reference 32 could be reference 34.
6- Line 76: Is it a typo the “x” in P. xcambivora ?
7- Line 82: “USA utilizes” by “USA, utilizes”.
8- Line 84: Check reference 34.
9- Line 169: A total of 576 soil samples is mentioned, however, previously is it cited that 40 sites were sampled with 7-10 samples per site. I got confused about numbers, it would be nice to revise them or to explain in a different manner.
10- Line 187: When mention “clean growing strains” you mean isolates from hyphal tips?
11- Line 223: Phytophthora in Italics?
12- Line 224: Phytophthora in Italics?
13- Line 241, 243: I was not able to find references: MK908981.1, 10838A1072, 3315A196, using BLAST. I wonder if there is any typo mistake with the codes, or they are not uploaded. Please, check also codes of page 10.
14- I suggest to upload your sequences to GeneBank, at least those for mt cox1.
15- Line 419: “Er. crassifolium” or “E. crassifolium”.
16- Regarding the references, I wonder if the scientific names should not be written in Italics and with capitals only for Genus initials. Please, check journal rules.
Author Response
Comments 1: The title could be modified in view of the origin of the samples as: “Phytophthora species and their associations with chaparral and oak woodland in Southern California”.
Response 1: We agree, thus title was modified as suggested.
Comments 2: Line 29: Please, consider to change “among the most” by “include species”.
Response 2: Sentence in line 34 was adjusted to "includes several species that are significant plant pathogens worldwide".
Comments 3: Line 39: Check ref 17 (is related to Vietnam (Asia)).
Response 3: Thank you for noticing, reference was updated to Duncan et al 2014 and Brasier et al. 2022.
Comments 4: Line 49: I wonder if is necessary to cite 6 references (31-36) for the information included in the paragraph.
Response 4: Each citation is in reference of each Mediterranean region in the world.
Comments 5: Line 69: Check if the reference 32 could be reference 34.
Response 5: Yes, reference was corrected with new numbering.
Comments 6: Line 76: Is it a typo the “x” in P. xcambivora ?
Response 6: it is not a typo. “x” is in reference to the hybrid nature of this species. Please revise: https://idtools.org/phytophthora/index.cfm?packageID=1131&entityID=5029
Comments 7: Line 82: “USA utilizes” by “USA, utilizes”.
Response 7:Modified as suggested. in line 87
Comments 8: Line 84: Check reference 34.
Response 8: Yes, reference was corrected with new numbering.
Comments 9: Line 169: A total of 576 soil samples is mentioned, however, previously is it cited that 40 sites were sampled with 7-10 samples per site. I got confused about numbers, it would be nice to revise them or to explain in a different manner.
Response 9:
Thank you for pointing this out. In total 40 locations were sampled, taking between 10 to 15 samples (number modified in manuscript) at each site, thus a total of 576 soil samples. Since it corresponded to an exploratory survey, with the main objective of determining Phytophthora diversity, sampling was biased towards plants that presented typical Phytophthora-associated symptoms (dieback, wilting or chlorosis). The same approach can be seen in other Phytophthora surveys conducted in different regions of the world (Scanu et al. 2015, Jung et al. 2018, Jung et al. 2020, and Jung et al. 2017.
Comments 10:Line 187: When mention “clean growing strains” you mean isolates from hyphal tips?
Response 10: It was in reference of pure cultures free of contaminants. Sentence in line 195 was modified to Pure cultures of each isolate were transferred to a sterile 1 ml pea broth solution and grown at 20 °C for at least 48 hours for DNA extraction.
Comments 11:Line 223: Phytophthora in Italics?
Response 11: Yes, updated.
Comments 12:Line 224: Phytophthora in Italics?
Response 12: Yes, updated.
Comments 13: Line 241, 243: I was not able to find references: MK908981.1, 10838A1072, 3315A196 using BLAST. I wonder if there is any typo mistake with the codes, or they are not uploaded. Please, check also codes of page 10.
Response 13:
MK908981.1 is available in genbank:https://www.ncbi.nlm.nih.gov/nuccore/MK908981.1/
10838A1072 is from (WPC) Worldwide Phytophthora collection from University of California Riverside and can be found here:https://microplantpath.ucr.edu/document/phytophthora-inventory-list. Some codes had not been updated in Genbank from which isolated were referenced in the manuscript. Corrected isolate number has been modified. This isolate also has a genbank accession: GU258767.1
3315A1962 is now referenced through CBS isolate number CBS:124696,
Non-GenBank codes in page 10 are in reference of local UC Davis Rizzo Lab culture collection. Clarifications were made in the manuscript in lines 262-267.
Comments 14: I suggest to upload your sequences to GeneBank, at least those for mt cox1.
Response 14: They have been uploaded since September 2024, i.e.: PQ216457.1 https://www.ncbi.nlm.nih.gov/nuccore/PQ216457.1/
Comments 15: Line 419: “Er. crassifolium” or “E. crassifolium”.
Response 15: “Er.” Was used for Eriodictyon crassifolium to avoid confusion when referencing the other plant species Eriogonum fasciculatum (E. fasciculatum) that share same first initial.
Comments 16: Regarding the references, I wonder if the scientific names should not be written in Italics and with capitals only for Genus initials. Please, check journal rules. Journal rules indicate that scientific names be in italics. In other manuscripts of JoF, scientific names are also italicized in references.
Response 16: Thank you for noticing this. After looking at other published articles in the JoF, both genus and species names are italicized in reference sections.
Reviewer 3 Report
The work is well designed and carried out properly and the results are well presented and discussed. The only issue I have with the study is that the authors did not do any pathogenicity test on the isolates they isolated from the native plant species. They related the signs/symptoms they observed on the plant species with Phytophthora spp. they isolated from. Some of these species have a wide-host range and have been reported to be pathogenic on other crops.
The text is well prepared, and tables and figures are well organized.
Author Response
Comments 1:
The work is well designed and carried out properly and the results are well presented and discussed. The only issue I have with the study is that the authors did not do any pathogenicity test on the isolates they isolated from the native plant species. They related the signs/symptoms they observed on the plant species with Phytophthora spp. they isolated from. Some of these species have a wide-host range and have been reported to be pathogenic on other crops.
Response 1:
Thank you for this comment. This is a very good point. We have done pathogenic tests on a subset of the recovered isolates on four chaparral species. However, after discussing with other co-authors, we decided that we will leave these pathogenic tests for a near future publication. Our main objective for the current manuscript was to highlight the encountered Phytophthora species diversity.
Reviewer 4 Report
Accept in present form
The oomycete genus Phytophthora, family Peronosporaceae, and kingdom Straminipila are among the most highly damaging plant pathogens globally. Fusarium proliferatum is one of the causes of the symptoms. In forests and natural ecosystems, invasive Phytophthora species can destabilize ecosystems with detrimental effects on biodiversity and on ecosystem services. Since the 1960s, there has been an exponential rise in devastating tree and shrub diseases caused by invasive Phytophthora species in natural ecosystems in Australia, Europe, and North America. California is one of the five main Mediterranean climate regions in the world. So, it is of great significance to study Phytophthora species and their association with chaparral vegetation. The results present the main outcomes and the potential risks associated with the presence of these Phytophthora species in the arid lands of Southern California. The article has appropriate and adequate references to related and previous work. The text is clear and easy to read and can be recommended for publication.
Author Response
Comment 1:
The oomycete genus Phytophthora, family Peronosporaceae, and kingdom Straminipila are among the most highly damaging plant pathogens globally. Fusarium proliferatum is one of the causes of the symptoms. In forests and natural ecosystems, invasive Phytophthora species can destabilize ecosystems with detrimental effects on biodiversity and on ecosystem services. Since the 1960s, there has been an exponential rise in devastating tree and shrub diseases caused by invasive Phytophthora species in natural ecosystems in Australia, Europe, and North America. California is one of the five main Mediterranean climate regions in the world. So, it is of great significance to study Phytophthora species and their association with chaparral vegetation. The results present the main outcomes and the potential risks associated with the presence of these Phytophthora species in the arid lands of Southern California. The article has appropriate and adequate references to related and previous work. The text is clear and easy to read and can be recommended for publication.
Response 1:
Thank you for these comments. No further modifications were made since no revisions were requested.
Round 2
Reviewer 1 Report
Dear authors,
Thank you for the revised manuscript. It was done well. The manuscript provides a comprehensive analysis of the Phytophthora species in Southern California, the tree and shrub diseases they cause, and their relationship with host plants. A large diversity of Phytophthora species was found beneath the surveyed plants. This study is significant, as it addresses the key link between understanding the increasing invasion of Phytophthora species and the interactions between soil and native plant communities, and for the first time detects and discovers new pathogenic mold taxa. A relatively robust method has been provided through soil sampling analysis, lesion isolation, quantitative sequencing, and phylogenetic inference. I have almost no issues for the MS.
1. The manuscript has undergone substantial improvements and there are no issues with its structure, rhetoric, or other aspects.
2. The research objectives of the abstract section have been clearly stated, and the background and importance have also been clearly explained.
3. In the methodology section, the bias of the sampling method is also clearly explained, which conforms to certain principles of random allocation and improves the reliability and generalization of experimental results.
4. In the conclusion section, This study narrowed the current knowledge gap of Phytophthora diversity in areas of Southern California, establishing new host associations, first detections and the discovery of novel Phytophthora taxa. the newly added content clarifies the research results and limitations, and also provides some ideas for future research ideas.